# AlphaFold Blindness to Topological Barriers Affects Its Ability to Correctly Predict Proteins’ Topology

**DOI:** 10.3390/molecules28227462

**Published:** 2023-11-07

**Authors:** Pawel Dabrowski-Tumanski, Andrzej Stasiak

**Affiliations:** 1Faculty of Mathematics and Natural Sciences, School of Exact Sciences, Cardinal Wyszynski University in Warsaw, Wóycickiego 1/3, 01-938 Warsaw, Poland; 2Center for Integrative Genomics, University of Lausanne, 1015 Lausanne, Switzerland; 3SIB Swiss Institute of Bioinformatics, 1015 Lausanne, Switzerland

**Keywords:** AlphaFold, protein structure prediction, topological barriers, knotted proteins, topology validation, residue gas model, overlapping residues

## Abstract

AlphaFold is a groundbreaking deep learning tool for protein structure prediction. It achieved remarkable accuracy in modeling many 3D structures while taking as the user input only the known amino acid sequence of proteins in question. Intriguingly though, in the early steps of each individual structure prediction procedure, AlphaFold does not respect topological barriers that, in real proteins, result from the reciprocal impermeability of polypeptide chains. This study aims to investigate how this failure to respect topological barriers affects AlphaFold predictions with respect to the topology of protein chains. We focus on such classes of proteins that, during their natural folding, reproducibly form the same knot type on their linear polypeptide chain, as revealed by their crystallographic analysis. We use partially artificial test constructs in which the mutual non-permeability of polypeptide chains should not permit the formation of complex composite knots during natural protein folding. We find that despite the formal impossibility that the protein folding process could produce such knots, AlphaFold predicts these proteins to form complex composite knots. Our study underscores the necessity for cautious interpretation and further validation of topological features in protein structures predicted by AlphaFold.

## 1. Introduction

AlphaFold [1,2,3] has revolutionized protein structure prediction, achieving unprecedented accuracy and surpassing other participants in two editions of the Critical Assessment of Structure Prediction (CASP) competition [4,5]. The quantitative leap was possible due to utilizing deep learning algorithms and large databases. This, in turn, fostered numerous modifications and implementations [6,7,8,9,10,11,12,13,14]. Following the success, the authors and developers of AlphaFold have created a broadly available, comprehensive database of predicted 3D protein structures, starting from the whole human proteome [15] and later expanding it to over 200 million entries offering a vast potential for exploration.

One of the subjects that can be studied in such a vast database is the existence of proteins with non-trivial topology. In fact, although a great majority of proteins fold into their native structure without forming knots on their polypeptide chains, there are also families of proteins whose polypeptide chains are reproducibly knotted during their folding [16,17,18,19]. In contrast to the entropically driven formation of various knots on long polymeric chains such as on long linear DNA molecules packed inside phage heads [20,21,22], the knots formed during protein folding are highly specific, where a given protein species always forms the same type of knot. In the majority of knotted proteins, it is a simple trefoil ($3_{1}$) knot with just three crossings in its minimal crossing diagram. However, the knots in proteins can also be more complex. For example, the polypeptide chain of bacterial α-haloacid dehalogenase always forms a complex knot with six crossings in its minimal crossing diagram [23]. The formed knot is known as Stevedore’s knot and has the topological notation $6_{1}$, once the two ends of the linear knots are connected with each other without introducing additional crossings. In addition to forming a unique type of knot characteristic for a given protein species, knotted portions of polypeptide chains take practically the same shape in each copy of a given knotted protein and can even be required for the formation of active sites of these proteins [24]. In total, about 2% of known structures are knotted, representing five different knot types [16,18,25]. It was therefore very tempting to search for new knotted families and knot topologies among the structures predicted by AlphaFold.

In this spirit, among the structures predicted by AlphaFold, Brems et al. identified two new knot topologies, with five and seven crossings in minimal crossing projection ($5_{1}$ and $7_{1}$ knots); the latter, if experimentally verified, would be the most complex protein knot to date [26]. Yet another knot type, a symmetric knot with six crossings ($6_{3}$ knot), was found in AlphaFold predictions by Perlinska et al. [27]. It is worth mentioning that all of those knots, if also identified in an experiment, would disprove the long-standing hypothesis that all protein knots are formed by single threading through a twist loop (so-called “twist knots”) [28]. The topology of all the structures predicted by AlphaFold was analyzed and gathered in the AlphaKnot database [29]. Yet, there remains the question of how precise AlphaFold predictions are in terms of topology. In fact, the topological analysis is a very delicate matter, as switching the position of two close chains may completely change the protein topology. To be aware of AlphaFold limitations, it is necessary to understand its algorithm.

In short, AlphaFold consists of two blocks. One, called Evoformer, builds an abstract protein representation from the protein sequence and its homologs. The second block is the structure module, the aim of which is to build a complete 3D structure starting from the abstract representation produced by Evoformer [1]. The abstract representation produced by Evoformer is a set of rotations and translations one has to perform to move each of the amino acids from the origin of coordinates to a desired place. The movement is carried out in the structure block. Remarkably, during adjustment of the structure, all the residues can move freely (suggestively called “residue gas”) and the peptide bond geometry is not conserved. The violation of geometrical constraints and chain connectivity is important from the viewpoint of the algorithm, as it greatly simplifies the calculations. The backbone geometry is corrected only in the last step, during fine-tuning of the structure within the Amber force field, the aim of which is only to remove stereochemical violations.

As a result, the continuity of the modeled polypeptide chain, which provides a basic topological constraint, is deliberately not maintained. This, in some specific cases, may negatively affect the algorithm’s predictions concerning the topology of polypeptide chains. In particular, when AlphFold predicts that a polypeptide chain of a given protein forms a knot of a given topological type, this may not reflect the reality, as some of the AlphaFold results may be impossible to realize during the real protein folding process.

Most of the knots observed in proteins are shallow, which means that the intrachain interlacing leading to the formation of knots is located very close to at least one of the ends of the polypeptide chain [19]. It is assumed that shallow protein knots form during the final stages of protein folding when the protein chain compacts into a globule and one or both of its termini interlace with the distally located portions of the same polypeptide chain. Alternatively, the loop can move with respect to the tail in a mouse-trap-like mechanism. Both versions of the mechanism were observed *in silico* in the case of proteins with shallow $3_{1}$ and $5_{2}$ knots [30,31,32,33].

Less frequent than shallow knots are deep knots, in which intrachain interlacing leads to the formation of the knot located further than ca 20 amino acids from the nearest termini of the knotted polypeptide chain [34]. Their formation requires threading of long portions of the chain through an e ncircling loop, which is not required for folding of shallow knots. It was shown that chain threading is the rate-limiting step, and the height of the energy barrier rises with the length of the chain needed to be pushed [32,35]. As a result, no simple threading was observed *in silico* as an efficient folding mechanism for deeply knotted proteins. In addition, no long-range threading of polypeptide chains was observed in *in silico* studies of protein folding. This observation is consistent with the notion that spontaneous protein folding, which needs no energy input, is unable to generate forces that can drive prolonged longitudinal motion of polypeptide chain portions through encircling loops in the same polypeptide chain. However, there are known biological mechanisms that can drive prolonged longitudinal motion of portions of polypeptide chains. During protein translation, ATP is used to induce the longitudinal motion of nascent segments of synthesized protein chains. Since protein folding occurs co-translationally, it was proposed that the longitudinal motion of nascent polypeptide chains of newly synthesized proteins may be instrumental in the process of knot formation during protein folding [36].

This proposal remained a theoretical possibility until the intriguing results of Mallam et al., who reported that the YibK protein, which normally folds into a trefoil knot, was still able to do so even when the protein was genetically engineered by adding two bulky, fast-folding and stable protein domains at both ends [37]. The authors called these terminal domains “plugs”, as they expected that these domains would act as plugs that would prevent the threading through the enclosing loop, required for knot formation. Such a threading is formally impossible if both ends of the protein form bulky plugs. However, during *in vitro* translation, without the aid of chaperones, the YibK variant with two plugs was still able to fold into trefoil knots, and this constituted a conundrum. This conundrum was naturally resolved though when considering knot formation by co-translational protein folding (see Figure 1).

In this mechanism, the twisted polypeptide loop is attached to the ribosome surface encircling the ribosome exit channel, and the nascent polypeptide chain is pushed through the encircling loop, eventually leading to the formation of the knot (Figure 1). Such a mechanism solves in principle the problem of folding of deeply knotted proteins, regardless of the length of the chain that has to be threaded, as the vectorial process of nascent polypeptide chain growth is in this case the driving force for threading. In particular, such a mechanism allows for the formation of a knot in YibK mutant with plugs, as the N-terminal plug can form as soon as it leaves the ribosome exit tunnel (Figure 1A), while the C-terminal portion of the nascent chain is pushed through the encircling, twisted loop, before the plug can be formed (Figure 1B,C).

An alternative mechanism for folding of deeply knotted proteins was proposed for structures composed of duplicated domains. In such a case, the domain swapping could lead to interlacing of the pieces of the chain, resulting in knotted conformation. Such a mechanism could, however, be valid only for proteins with a symmetrical knot, such as a deep $4_{1}$ knot present in Acetohydroxy acid isomeroreductase (PDB code 1yve) [39]. Moreover, this mechanism does not explain how the terminal plugs would not block the process of knotting.

Taking all known mechanisms into account, one can analyze what types of knots are biologically accessible. In this work, we construct a series of examples of artificial protein structures that are inaccessible because of kinetic and topological reasons and still were predicted to be knotted by AlphaFold. As a result, we suggest treating the AlphaFold results with caution, especially when investigating topological aspects of protein structures.

## 2. Results

### 2.1. Alphafold Builds Protein Structures Forming Arbitrary Complex Knots

In the introduction, we discussed currently known mechanisms that possibly lead to the formation of protein knots: threading, or a mouse-trap mechanism for shallowly knotted proteins, and co-translational folding or domain duplication for deeply knotted proteins (Figure 2). The on-ribosome folding can produce only a single deep knot, as each knot requires a twisted loop surrounding the ribosome exit tunnel. It is highly doubtful that two such loops would attach to the ribosome surface in the desired conformation. Similarly, domain swapping results in a single knot, as it is an outcome of swapping the termini position. Threading the tail could in principle lead to two consecutive knotted domains (as there are two ends that can be threaded—Figure 2A).

Formation of a higher number of consecutive deep knots would require passing very long tails or already-folded, knotted domains through the twisted loop, which again seems doubtful. Eventually, the combination of on-ribosome folding (Figure 2B) or domain swapping with tail threading could lead to a maximum of three consecutive knots (Figure 2C)—two shallow knots on the termini and a deep knot in the center.

We, therefore, tested the ability of AlphaFold to predict the folded structure of an artificial protein in which the polypeptide chain is composed of multiple tandem repeats of polypeptide chains of naturally knotted proteins. We started with the smallest knotted protein known—MJ0366 from *Methanocaldococcus jannaschii* (PDB code: 2efv)—which is known to form a shallow trefoil knot. Interestingly, AlphaFold predicted that the structure of multiple repeats of the MJ0366 sequence forms a composite knot, where each of the MJ0366 domains forms a trefoil knot (see Figure 3). For practical reasons (computer memory requirements and calculation time), we limited our tests to polypeptide chains with up to 10 tandem repeats, but it seems that constructs with any arbitrary number of MJ0366 repeats would be predicted by AlphaFold to form proteins with the corresponding number of linearly arranged trefoil-forming polypeptide blocks (Figure 3).

It needs to be mentioned here that in its native form, the polypeptide chain of a single domain of MJ0366 from *Methanocaldococcus jannaschii* forms a shallow trefoil knot, which is expected to be formed by a simple, shallow threading [32]. On the other hand, in the artificial construct tested here, only the two terminal repeats could form shallow knots by the interlacing of their free ends with some accessible polypeptide chain portions of the same protein. The inner domains should either remain unknotted or form a series of slipknots, as the chain is evolutionarily optimized for shallow threading. Formation of deep knots would require threading long pieces of the chain (the whole domains), which would in turn require a change in the folding mechanism. However, in contrast to the expectation, the structure predicted by AlphaFold for this tandem repeat of MJ0366 is not composed of multiple slip knot-forming portions but of portions forming complete knots. Obtaining such a structure would require threading a few domains through the twisted loop, which was not observed so far even for natively deeply knotted proteins.

We also tested AlphaFold predictions for artificial structures composed of multiple repeats of amino acid blocks that in a natural setting form deep knots in polypeptide chains. We used trimeric (3x) and pentameric (5x) repeats of amino acid sequences taken from the protein YibK (PDB code: 1j85), a t-RNA methyltransferase from *Haemophilus influenzae Rd KW20*, as it is the most commonly studied knotted protein by both simulations and experiments. This protein naturally forms a deep trefoil knot. As shown in Figure 4, AlphaFold predicted that these constructs will fold into forms with three and five knots, respectively, which again suggests that it can produce a chain with an arbitrary number of consecutive knots. Again, such structures are impossible to form using the known mechanisms of knot formation in proteins.

The full list of protein models and their topology may be found in Appendix A. In our study, the domains were usually connected with a flexible, nine-glycine linker, to remove the possible influence of the linker on the domain structure. However, for the completeness of the study, we also tested various linker types (using glycine-serine linkers and proline-reach linkers) and lengths (in the range of 1–17 residues). Apart from the conformational variability of linkers, we always obtained topologically equivalent results (see Appendix A).

### 2.2. Alphafold Predicts Impossibly Densely Packed Structures

The structures presented before were multidomain chains. We also considered whether it is possible to obtain a falsely knotted, single-domain structure. In order to produce one, we started again from the YibK protein, which naturally forms a trefoil knot. We have, however, shortened the twisted loop through which the terminal portion of the chain would need to thread to form a knot. If the twisted loop is too short, the threading should become impossible and knots should not form or the loop should change its conformation to adapt to the bulky threading chain.

However, even after the removal of eight (around 30%) residues from the twisted loop (twenty-six residues delimited by β-strands and spanning residues Arg74-Phe99), AlphaFold still predicted a knotted structure with no significant conformational change apart from the missing part of the loop (Figure 5A). In addition, we were able to exchange small residues for bulky tryptophans, producing an extremely dense packing within the twisted loop pierced by the threaded portion of the knotted polypeptide chain (for sequences used, see Appendix A). Although no stereochemical clashes were observed within the twisted loop, any movement needed for threading seems impossible (Figure 5B).

## 3. Discussion

In this study, we proposed a series of structures, which according to the currently known mechanisms of protein knotting, should have a different topology than the one proposed by AlphaFold. Until the experimentally determined structure is known, we cannot definitely say that AlphaFold predicts the topology incorrectly, especially as there may be other unknown mechanisms for protein knotting. Yet, our results (protein with 10 consecutive knots) and the AlphaFold algorithm (moving the residues without preserving the chain connectivity) indicate that even if some structures presented are actually plausible, AlphaFold still cannot be trusted in terms of protein topology.

The failure to respect the topology is a very deeply rooted feature of the algorithm, as it is required to treat the chain in a reduced form of rotations and translations predicted by one of the AlphaFold modules. Therefore, fixing the topological problem would require changing the deep learning architecture of the tool. As a result, additional validation of predicted topologies is imperative in any studies of AlphaFold-predicted structures, where the topology is important.

Most importantly, such validation has to be motivated biologically. Assessing the final structure using any local metric seems to be insufficient. In particular, the AlphaFold algorithm relies on the optimization of a metric called pLDDT (a version of lDDT—Local Distance Difference Test metric [40]). The metric calculates how optimal inter-residue distances are, where the optimum is taken from known, homologous structures. The metric attains values in the range of 0–100, where values larger than 70 are treated as “a generally good backbone prediction”, while those with pLDDT higher than 90 are “modeled to high accuracy”. The metric is local in the sense that it is calculated for each residue individually; however, usually, the mean value is presented to assess the structure, which may be insufficient for judging the topology, as it depends on local chain placement. It has to be noted that all of the structures analyzed in this work have mean pLDDT larger than 76, and in most cases larger than 80, and therefore, they are falsely classified as modeled reasonably. This stems from the fact that in the case of knotted structures, crucial in structure validation is the mutual location of the loop and threaded chain—information which, being local, is easily lost while averaging the pLDDT metric over the whole chain.

However, calculating the per-residue pLDDT does not solve the problem. In fact, in our cases, the lowest pLDDT is obtained for the linkers, while for each knotted core, the pLDDT is always relatively large (Figure 6). This is, however, expected, as pLDDT measures how optimal the distances are, where the optimum is taken from the reference structures. Therefore, the more similar each knotted core is to the native structure, the higher the pLDDT score.

This is the general problem with local measures such as pLDDT. Locally, each domain and knotted core has the correct structure. Therefore, to fully assess the predicted structure, some global metric, taking into account the whole backbone at once, should be used. One such metric may be the global topology of the protein. The topology may be characterized by various knot invariants, such as Alexander or Jones polynomial, which can be obtained for example directly on the KnotProt webserver [19], or with a scripting approach using the Topoly Python package [41]. However, one has to know the expected topology of the whole chain. Again, such information may be motivated biologically, assuming the possible folding mechanism of a protein of interest.

There is also another problem with local metrics such as pLDDT, which may be seen in multidomain proteins. In the case of both proteins (MJ0366 and YibK), there are crystal structures containing homodimers. The spatial proximity of the domains in such structures is passed through the AlphaFold algorithm, and in final structures, the outcoming domains commonly exist in close pairs (see, e.g., Figure 3B, where red and orange domains are spatially close, or Appendix A). However, if there are, e.g., three tandem repeats of a domain, the highest value of the pLDDT metric would be obtained when the terminal domains were overlapped. Only then would the mutual location of each terminal domain relative to the central part reflect the one from the homodimeric crystal structure. As AlphaFold permits chain passing, structures with overlayed domains are possible. In fact, although we did not find such structures in the best-ranked AlphaFold models, we indeed found such trimers with overlayed domains among models with a lower ranking (Figure 7). It is worth mentioning that such structures also have mean pLDDT larger than 70, so they should be regarded as reliably modeled. Again, such structures may be easily eliminated by applying a global characteristic, such as the topology of the whole chain. In the case of overlayed domains, the resulting topology would be impossible to realize.

The analysis of tandem repeats of knotted proteins are examples of (most probably) false positive results of knotting performed by AlphaFold. As a control, one may also study the true positive and true negative examples. The best case would be a pair of close mutants differing in the topology. However, it was shown that the topology is highly conserved and does not change even after a large change in the sequence [42]. The best example of a pair of similar proteins differing in topology is N-acetyl-L-ornithine transcarbamoylase (ATC—$3_{1}$-knotted) and ornithine transcarbamoylase (OTC—unknotted), with 30% sequence identity (measured by the jFatCat algorithm, present in the RCSB database [43,44]). According to the expectation, AlphaFold predicts the structure and the topology well (see Appendix A). This is the consequence of a vast database of sequences and structures known to AlphaFold. In fact, even when manually removing the sequences of homologs closest to the target, AlphaFold is still able to identify the templates with the correct topology, eventually modeling the ATC/OTC pair well.

The experiment with the tight loop in the YibK artificial mutant showed another problem with AlphaFold, that small changes in the sequence, although potentially having a drastic effect on the topology, have only limited influence on the structure predicted by AlphaFold. This effect was studied before in the case of missense mutations [45,46], but it shows that AlphaFold is of limited use in the case of lasso-like proteins [47,48], where the topology can change when introducing or removing single-cysteine residue.

The whole analysis shows that AlphaFold, by design allowing chain passage, is not suited to model structures with topological features such as knots. Yet, some knotted structures identified among AlphaFold results may still be real. The $5_{1}$ and $7_{1}$ knots identified by Brems et al. [26] are single-domain structures with no visible dense residue packing. The $6_{3}$ knot found by Perlinska et al. [27] is a double-domain knot; however, the domains are swapped. A similar effect was already observed in the analysis of a possible evolutionary pathway of the deep $4_{1}$ knot, another topologically symmetric structure [28], which opens the pathway for the existence of the $6_{3}$-knotted protein.

In general, however, caution should be exercised when interpreting AlphaFold’s predictions in terms of protein topology, considering the potential for false positive identification of knots and other topological features.

When this work was undergoing a revision in reply to the comments of the referees, a new paper was published on a closely related subject [49]. Doyle et al., in their article entitled “De novo design of knotted tandem repeat proteins”, have shown, using crystallographic analysis, that proteins whose amino acid sequence was designed to fold into complex knots were unable to fold into such complex knots. AlphaFold, however, predicted the successful folding of these proteins into complex knots. Doyle et al. invoked topological barriers as obstacles in the folding of these proteins into the knotted structure predicted by AlphaFold.

## 4. Materials and Methods

### 4.1. Software and Hardware

We utilized AlphaFold 2.3.2 as a Docker image on the Google Cloud Platform, employing n1-highmem-8 instances with one NVIDIA Tesla K80 GPU. The Life Sciences API and Cloud Shell facilitated job execution, with output saved in a dedicated Google Cloud Storage bucket. All necessary scripts are available in the GitLab project repository (https://gitlab.com/pdabrowskitumanski/alphafoldwrapper (accessed on 27 October 2023)). A full version of BFD and the *monomer* protocol of AlphaFold were utilized when running the jobs. For each sequence, AlphaFold proposed the 5 best models. If not stated otherwise, the analysis applies to the best model. All the models obtained, as well as their topology, are present in the GitLab repository.

### 4.2. Structures and Sequences

Amino acid sequences testing AlphaFold predictions of protein structures were downloaded from the RCSB database or modified accordingly. A detailed list of the sequences used, along with the corresponding models obtained and their topology, is available in the GitLab project repository. UCSF Chimera [50] was employed for structure analysis and visualization. All the sequences along with their observed topology can be found in the GitLab repository (https://gitlab.com/pdabrowskitumanski/alphafoldwrapper (accessed on 27 October 2023)).

## Figures and Tables

**Figure 1 molecules-28-07462-f001:**
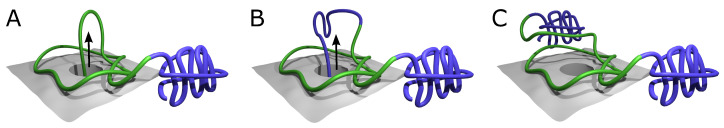
**Ribosome-based co-translational folding mechanism** [38]. The mechanism allows the formation of a deeply knotted protein with two “plug” domains on both termini. (**A**) Part of the central domain chain (green) surrounds the ribosome exit tunnel, forming the twisted loop attached to the ribosome surface (gray). The loop is threaded by the nascent C-tail going out of the ribosome. The N-tail may freely fold into the N-terminal domain, e.g., bulky plug (violet part of the chain). (**B**) The chain is threaded through the loop and may start the formation of the C-terminal domain (violet part of the chain). (**C**) Only after the chain is fully formed and pushed through the twisted loop may it form the C-terminal bulky domain (violet chain in the background). In this state, the chain is already knotted with two fast-folding bulky domains formed. After detachment of the loop from the ribosome surface, the central domain may fold into the native structure. The arrows indicate the movement of the chain out of the ribosome exit channel.

**Figure 2 molecules-28-07462-f002:**
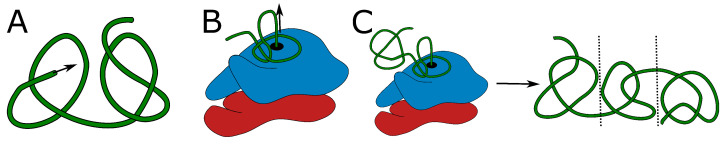
**Proposed mechanisms of knot formation.** (**A**) Direct threading of the tail can produce up to two knots, most probably shallow. (**B**) On-ribosome knotting requires attaching the loop to the ribosome surface and therefore allows the formation of a single deep knot [38]. (**C**) The composition of threading and on-ribosome folding allows the creation of three consecutive knots (separated by dashed lines)—one deep knot in the center surrounded by up to two, most probably shallow knots formed by the termini. The colors in panels (**B**,**C**) show large and small subunits of a ribosome. Green is the protein chain. The arrows indicate the chain movement leading to a knot.

**Figure 3 molecules-28-07462-f003:**
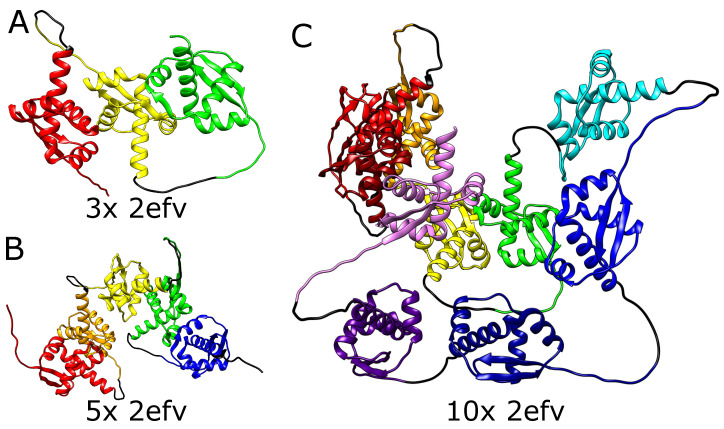
**Structures of MJ0366 (PDB code 2efv) multimers.** (**A**) Trimer with 3 consecutive knots, (**B**) pentamer with 5 consecutive knots, (**C**) decamer with 10 consecutive knots. In each panel, the domains are depicted in different colors. Black strands denote the glycine linkers.

**Figure 4 molecules-28-07462-f004:**
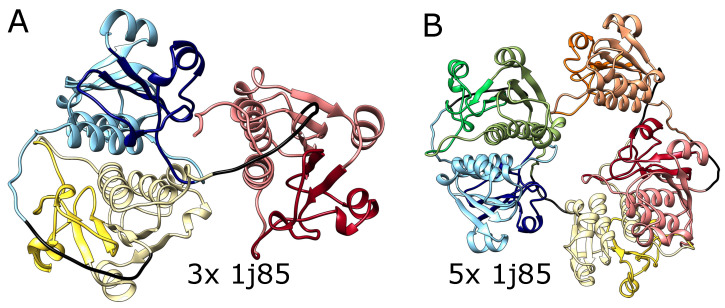
**The structures of multimers of deeply knotted YibK protein (PBD code 1j85)**. (**A**) Trimer with 3 consecutive trefoil knots, (**B**) pentamer with 5 consecutive trefoil knots. In both panels, the tandemly repeated protein blocks are represented with different pastel colors. The darker colors indicate the knotted core. The glycine linkers are presented in black.

**Figure 5 molecules-28-07462-f005:**
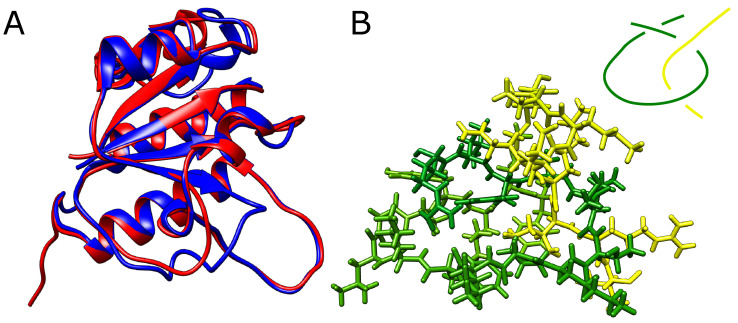
**Modified YibK protein with shortened loop.** (**A**) The native (blue) structure of YibK overlayed with the structure with 40% of residues removed from the twisted loop (red). The structures differ only in the region of the modified loop. (**B**) The loop (green) with the threaded tail (yellow) with all the atoms explicitly shown. In the top left corner is shown a schematic depiction of the threading. The native twisted loop is delimited by green β-strands and spans indices Arg74-Phe99.

**Figure 6 molecules-28-07462-f006:**
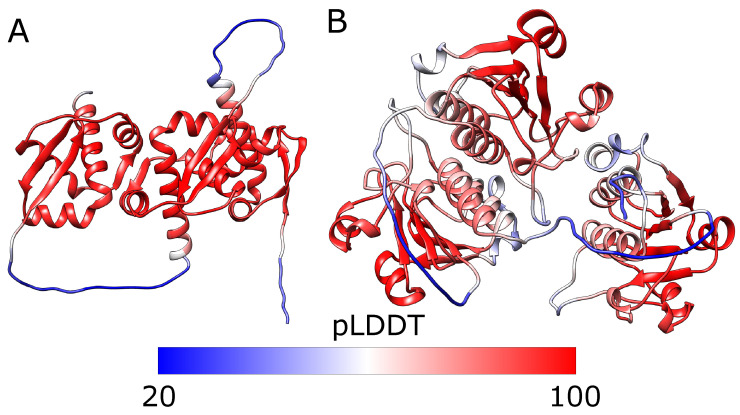
**The predicted structures of tandem triple repeats colored by pLDDT.** (**A**) 2efv trimer and (**B**) 1j85 trimer (right). Both structures have three consecutive trefoil knots. The lowest pLDDT (blue) can be seen in the linkers which are flexible and do not have homologs with well-defined structures. The knotted cores’ pLDDT is relatively high (red), indicating that those regions are modeled reliably. The white parts are the tails with medium values of pLDDT. Below the structures is the scale bar.

**Figure 7 molecules-28-07462-f007:**
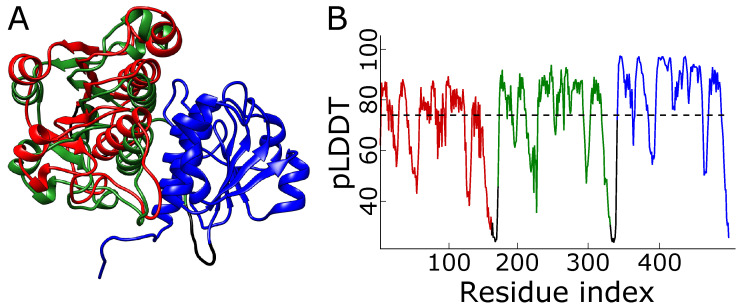
**Wrong structure with overlayed domains.** (**A**) The model proposed by AlphaFold for the 1j85 trimeric repeat ranked in second place. The terminal domains—red and green—are almost overlayed. (**B**) The plot of pLDDT for the structure. The dashed line denotes the mean pLDDT score, equal to 74. The colors in panel (**A**) match those in panel (**B**).

## Data Availability

Initial data (protein sequences and 3D structures) can be found in the RCSB database. Obtained results can be found in the GitLab repository https://gitlab.com/pdabrowskitumanski/alphafoldwrapper (accessed on 27 October 2023).

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
