# Peer review of "AlphaFold Blindness to Topological Barriers Affects Its Ability to Correctly Predict Proteins’ Topology"

_molecules, 2023, doi:10.3390/molecules28227462_

Round 1
Reviewer 1 Report
Comments and Suggestions for Authors
This study aims to assess the ability of AlphaFold to predict the complex folding topology of proteins chains, with a focus on the complex knots. To this end, the authors used two types of sequences: multidomain proteins with composite knots, and a single-domain proteins with modified knotted core. They reported that AlphaFold failed to predict topologically valid structures, primarily due to not considering topological barriers in the early steps of the modeling procedure. The manuscript is concise and provides some interesting results for general readers.
However, the reviewer still has some concern about the feasibility of the logic within their research. The authors built artificial tandem repeats from the known knot protein, and reported that AlphaFold predicted complex knots that natural protein folding can not form, based on known mechanisms of knot formation. However, the judgment is just based on inference instead of any experimental evidence. In fact, we have no idea about how these artificial tandem repeats will really fold without experimental structure. In short, the authors used an artificial sequence, and reported the software failed to predict it in a natural way. I’m wondering whether such a design is logically sound. I wish the authors could make explanations to the reviewer or within the discussion part of the manuscript.
Author Response
We thank the reviewer for the work and the comments. Below we gathered the point-by-point response to all remarks.
This study aims to assess the ability of AlphaFold to predict the complex folding topology of proteins chains, with a focus on the complex knots. To this end, the authors used two types of sequences: multidomain proteins with composite knots, and a single-domain proteins with modified knotted core. They reported that AlphaFold failed to predict topologically valid structures, primarily due to not considering topological barriers in the early steps of the modeling procedure. The manuscript is concise and provides some interesting results for general readers.
We thank the reviewer for this nice summary of our manuscript and for kindly stating that our manuscript provides some interesting results for general readers.
However, the reviewer still has some concern about the feasibility of the logic within their research. The authors built artificial tandem repeats from the known knot protein, and reported that AlphaFold predicted complex knots that natural protein folding can not form, based on known mechanisms of knot formation. However, the judgment is just based on inference instead of any experimental evidence. In fact, we have no idea about how these artificial tandem repeats will really fold without experimental structure. In short, the authors used an artificial sequence, and reported the software failed to predict it in a natural way. I’m wondering whether such a design is logically sound. I wish the authors could make explanations to the reviewer or within the discussion part of the manuscript.
We based our inference on the current knowledge about current knowledge about mechanism of knotted protein folding. However, according to the request of the referee, we have stated now in the discussion that there can be some unknown yet mechanisms by which such complex protein knots can form. We also pointed out clearly, that without having experimentally determined structure of protein constructs presented in our MS there is still a possibility that AlphaFold predictions my hold for these protein constructs.
Reviewer 2 Report
Comments and Suggestions for Authors
Please see pdf file Tumanski_Review for comments to authors

The paper is well written with only minor spelling and grammatical typos.
Author Response
We thank the reviewer for all the valuable comments, which helped us improve our manuscript. We included the response to the reviewer's comments in the attached PDF file.

Reviewer 3 Report
Comments and Suggestions for Authors
The article raises an interesting problem on the limits of Alphafold. The authors present examples based on the structural predictions of artificial proteins. They show that AlphaFold is able to predict models with unrealistic topologies for artificial proteins with succession of knotted modules or a modified knot protein with a loop too short for forming a knot. Nevertheless, it is difficult for me to understand what the real point is for the authors.
AlphaFold is not magic. It is based on analysis of sequences and structures and performs homology modelling. It is more sophisticated than other homology modelling programs but predictions are based on PDB structures, not on topology. AlphaFold predicts models it is asked to build. If AlphaFold is asked to build a topologically wrong molecule, it will do it. It might be possible to modify the algorithm to add warnings, but the model cannot be topologically right if the sequence does not permit it.
Here some suggestions to improve the paper:
- The authors should compare predictions made with AlphaFold and “manual” predictions made with MODELLER, to show the gain, if any, with AlphaFold.
- Can the authors provide an example of proteins made of a succession of knotted modules or of slipknots?
- The alignment of wild type and modified YibK proteins should be shown.
- The multimer structures should be compared to the dimeric structures of the templates indicated in the PDB, since 2efv and 1j85 crystallized as dimers. Is this information used by AlphaFold ?
- The problem of unrealistic structures not detected by a single metric is recurrent in molecular modelling. The authors should compare several metrics, and indicate a metric able to detect structures with overlayed domains. They should warn on the necessity of visual controls!
- Finally, the authors should propose a metric based on topology for a posteriory control of the topological quality of the models.
Author Response
We thank the reviewer for the valuable comments, which definitely helped us improve our manuscript. We attach the PDF with the full response to the reviewer's comments.

Round 2
Reviewer 1 Report
Comments and Suggestions for Authors
The response and revisions have solved my concerns.
Author Response
We again thank the reviewer for the work done with our manuscript which helped us enhance its value.
Reviewer 2 Report
Comments and Suggestions for Authors
there is one small typo in figure 1 legend - „plug”
the first quote is written as subscript but should be superscript for plug.
Author Response
We again thank the reviewer for many valuable comments and for spotting the LaTeX-based oversight. It has been fixed according to the reviewer's comment.